

# Characteristics of the vaginal microbiome in cross-border female sex workers in China: a case-control study

Xiang Hong[1], Shenghao Fang[2], Kaiping Huang[1], Jiechen Yin[1], Jianshuang Chen[1], Yan Xuan[1], Jing Zhu[1], Jun Ma[1], Pengfei Qin[3], Danhong Peng[3], Ning Wang[4] and Bei Wang[1]

[1] Key Laboratory of Environmental Medicine and Engineering of Ministry of Education, Department of Epidemiology and Health Statistics, School of Public Health, Southeast University, Nanjing, China
[2] Mount Sinai Health System, New York, NY, USA
[3] Department of Obstetrics and Gynecology, Zhong Da Hospital, School of Medicine, Southeast University, Nanjing, China
[4] National Center for AIDS/STD Control and Prevention, Chinese Center for Disease Control and Prevention, Beijing, China

Corresponding author
Bei Wang, wangbeilxb@163.com

## ABSTRACT

**Background:** Female sex workers (FSWs) are key groups in the transmission of sexual transmitted infections (STI), and vaginal microbiome variations play an important role in transmission. We aimed to explore the characteristics of vaginal microbiome among FSWs.

**Materials and Methods:** A total of 24 cross-border FSWs were randomly selected from a cross-sectional survey for female sex workers in southwest China. Thirty-seven female non-sex workers (FNSWs) were randomly selected from the gynecology clinic and health examination center. Vaginal swabs were collected, bacterial DNA extracted and 16S rRNA genes were sequenced. Differences in the vaginal microbiome between both groups were compared using bioinformatics analysis.

**Results:** One DNA sample was excluded due to unqualified concentration, therefore 60 samples were sequenced. FSWs had significantly different vaginal microbiota β diversity, but undifferentiated α diversity when compared with non-sex workers. The average relative abundance of *Sneathia*, *Shigella*, *Neisseria*, *Chlamydia*, *Prevotella*, *Enterococcus* and *Ureaplasma* among FSWs was higher than FNSWs, and relative abundance of *Atopobium* in FSWs was lower than FNSWs. The *Lactobacillus* genus was the major genus in both groups. At the species level, *Lactobacllus crispatus*, *Lactobacllus gasseri* and *Lactobacllus jensenii*, in female sex workers, were lower when compared to FNSWs.

**Conclusion:** There were distinct differences in vaginal bacteria variety between FSWs and FNSWs. Some disease-related genus were also more abundant in FSWs. Based on these observations, further research is required to identify microbiome communities related to high STI risks and other diseases in these cohorts.

## INTRODUCTION

The current evidence suggests that female sex workers (FSWs) are key factors in the transmission of human immunodeficiency virus (HIV) and other sexually transmitted infections (STIs) (*Ferreira-Junior et al., 2018*). Every year, STIs affect more than 500 million people worldwide and lead to large financial burdens in hospitals, communities and health systems (*Gottlieb et al., 2014*). Although most STIs are curable with timely diagnosis and treatment, others may lead to serious long-term effects, including reproductive complications and death, especially among FSWs who are usually clandestine and do not have access to medical support (*Su et al., 2016*). Recent epidemiological investigations have observed unique characteristics in FSWs in the border region of Yunnan province, China, where the vast majority of FSWs come from Vietnam (*Wang et al., 2012*; *Zhu et al., 2018*). It is believed that these cross-border FSWs play key roles in STI transmission, because of frequent travelling and their tendencies to hide work histories (*Zhu et al., 2018*). The prevalence of STIs and related complications among the general population in local areas is difficult to control. The cross-border sex industry is a complicated social problem; however, it is important to understand and recognize STI or HIV risk factors to public health.

The vaginal microenvironment is a vital factor for STI transmission (*Van de Wijgert, 2017*). In general, *Lactobacillus* is the predominant bacteria in the vaginal flora. The *L. crispatus* was always regarded as a beneficial bacterium, which breaks down glycogen to produce lactic acid, hydrogen peroxide and other anti-microbial factors to balance the vaginal microflora (*Ravel et al., 2011*). Nevertheless, the role of *L. iners* in the vaginal microbiome was unclear (*Petrova et al., 2017*). The vaginal microbiome was usually a low-diversity, *lactobacilli*-dominated community; otherwise, it would be considered as vaginal dysbiosis (*Van de Wijgert, 2017*), where the mucosal barrier of vaginal was easier to be disrupted. Hence, the risk of STI infection is higher in women with vaginal dysbiosis (*Ziklo et al., 2018*). Similarly, cervico-vaginal inflammation is likely to increase the concentration of lymphocytes, which were target cells for HIV at mucosal sites. When HIV exposure occurs, the infection rates become increased (*Selhorst et al., 2017*). Meanwhile, the increased relative abundance of some opportunistic pathogens in the vagina, such as *Gardnerella vaginalis* and *Mycoplasma*, are also associated with bacterial vaginitis (BV) (*Onderdonk, Delaney & Fichorova, 2016*), endometriosis (*Campos et al., 2018*), infertility (*Tsevat et al., 2017*) and adverse pregnancy outcomes (*Murtha & Edwards, 2014*).

With the development of next generation sequencing, researchers have focused on the vaginal microbiome structure and composition effects on human health (*Chen et al., 2017*), and not only single bacterial strains. When acquiring the abundance information of all species, the vaginal microbiota community could be classified as different community state types (CSTs) by some mathematic algorithms. The most common type of classification is a five-state model, each of which is characterized by a specific and typical composition and an abundance of taxa: CST I is dominated by *L. crispatus*, CST II is dominated by *L. gasseri*, CST III is dominated by *L. iners*, CST IV is highly diverse and CST V is dominated by *L. jensenii* (*Lewis, Bernstein & Aral, 2017*). Further research has shown that different CSTs

have different effects on human health, and that different *Lactobacillus* species do not have the same benefits (*Petrova et al., 2017*). However, evidence showing the unique characteristics of vaginal microbiome in FSWs is still lacking. Notwithstanding this fact, *Wessels et al. (2017)* used 16S high-throughput sequencing technology to report that high-risk sexual behavior in Kenyans was associated with vaginal microbiota diversity and a lack of *Lactobacillus*. In considering that the vaginal microbiome was diverse with different ethnic backgrounds (*Ravel et al., 2011*), the characteristics of vaginal microbiome among Asian FSWs is worthy of study.

Here, we conducted a case control study to determine differences in the vaginal microbiome between FSWs and female non-sexual workers (FNSWs) in a border region of China. We hoped to understand the possible effects of commercial sex behaviors on the vaginal microbiome, and provide some new thoughts for FSWs management.

## MATERIALS AND METHODS

### Study participants

Vaginal swabs from FSWs were collected in a FSW cross-sectional study conducted by the Chinese Center for Disease Control (CDC) from June to December 2015 in Hekou County, Yunnan Province, China. Elements of this program have been reported in previous studies (*Reilly et al., 2012*; *Wang et al., 2015*; *Wang et al., 2012*; *Zhu et al., 2018*). All eligible FSWs were over 16 years old and had reported they had provided sexual services in exchange for money within the past 6 months. Sexual working conditions, smoking status, pregnancy and vaginitis history were self-reported. Some Vietnamese FSWs who could not speak Chinese were interviewed by a translator. For the collection of vaginal swabs, we excluded FSWs who were sexually active, or who had taken antibiotics in the previous 12 h, or were menstruating. The specific power and sample size estimation of microbiome studies (*Kelly et al., 2015*) pointed that 20 subjects per group allows more than 90% power to detect an $\omega^2$ of 0.03. Therefore, we selected 24 HIV-negative FSWs by random for sequencing analysis (12 Vietnamese and 12 Chinese women). Because we could not completely distinguish the FSW or NFSW women in the local hospital for the high proportion of FSWs there, and because they might not be willing to tell us their real careers, we recruited 37 FNSWs from the gynecology clinic and health examination center in Zhongda hospital, Nanjing city, another southern city of China, to reduce the potential misclassification bias. Demographic characteristics were collected along with vaginal swabs. One FSW swab was removed because of too low nucleic acid concentration. The CDC granted Ethical approval to carry out the study (Ethical Application Ref: X120331209), and written informed consent was obtained from all participants in the study.

### Vaginal swabs collection

All participants were placed in a lithotomy position for swab collection. A gynecologist swabbed for vaginal secretions and then examined the vagina with a speculum. A dry sterile swab was used to scrub secretions at the posterior fornix by rotating three times. The swabs were kept in a dry tube with a unique identification number and immediately placed in a 4 °C sampling collection box. They were then transferred to a −80 °C

refrigerator to await nucleic acid extraction. The samples were transported to the laboratory in drikold.

## DNA extraction and 16S rRNA gene sequencing

Swabs were thawed and placed in one ml PBS buffer solution. After 10 min oscillate (Water-bathing Constant Temperature Vibrator, Guohua Machinery Co., Ltd., Jiangsu, China), swabs were removed and the PBS eluates centrifuged for 2 min (12,000 rpm, Centrifuge 5424 R; Eppendorf Co., Ltd., Hamburg, Germany). Sediments were DNA extracted using TIANamp Bacteria DNA Kit (Tiangen Biochemical Technology, Beijing, China), following manufacturer's instructions. Purified nucleic acids were eluted in TE buffer and quantified on a Nanodrop 2000 (Thermo Fisher Scientific). As stated, one DNA sample of unqualified concentration (<10 ng/μl with an A260/280 ratio outside 1.6–2.0) was removed.

The hyper-variable V3-V4 region of the 16S rRNA gene was amplified by PCR, using modified 338F and 806R primers with a unique 12 bp barcode, facilitating sequencing on the Illumina HiSeq platform (Illumina, San Diego, CA, USA). Primers were synthesized by Invitrogen (Invitrogen, Carlsbad, CA, USA). PCR reactions, containing 25 μl 2× Premix Taq (Takara Biotechnology, Dalian Co., Ltd., China), one μl each primer (10 mM) and three μl DNA (20 ng/μl) template in a final volume of 50 μl, were amplified using the following parameters: Initialization for 5 min at 94 °C; 30 cycles of 30 s denaturation at 94 °C, 30 s annealing at 52 °C, and 30 s extension at 72 °C; followed by a 10 min elongation at 72 °C. PCR was performed on a BioRad S1000 (Bio-Rad Laboratory, CA, USA). The PCR products were sequenced on an IlluminaHiseq2500 platform.

## Data processing and statistical analysis

Quality filtering on the paired-end raw reads was performed under specific filtering conditions to obtain high-quality clean reads according to the Trimmomatic (V0.33) quality-controlled process. Paired-end clean reads were merged using FLASH(V1.2.7), according to the relationship of the overlap between the paired-end reads. Sequences with ≥97% similarity were assigned to the same operational taxonomic units (OTUs) using the QIIME software (version 1.8.0). For each representative sequence, the silva database (https://www.arb-silva.de/) was used to annotate taxonomic information. OTU abundance information were normalized using a standard of sequence numbers corresponding to the sample with the least sequences. All bioinformatics analyses were completed on the Biomarker biocloud platform (https://www.biocloud.net).

Mothur software (version v.1.30) assessed alpha diversity, including ACE, Chao1, and the Simpson and Shannon index (*Schloss et al., 2009*). Principle Coordinate Analyses (PCoAs) based on Binary_jaccard dissimilarity, permutational multivariate analysis of variance (PERMANOVA) and cluster heatmap were generated to statistically compare microbial populations (β diversity) between groups. Euclidean distances between different samples or genera were calculated, and the *z*-score transformation for relative abundance of specific genus was generated to create the cluster heatmap. To illustrate relationships between some environmental factors and microbiota communities, a redundancy analysis

(RDA) was performed. A Linear Discriminant Analysis (LDA) Effect Size (Lefse), a Random Forest analysis and a multivariate Kruskal–Wallis test ($P$ was adjusted by the BH-FDR method) were performed to compare the relative abundance of specific genus between groups. These statistical methods were performed on the Biomarker biocloud platform using related R packages. To identify specific *Lactobacillus* species, representative sequences were queried against NCBI's 16S rRNA gene database using BLAST. The highest scoring species (>97% identity and coverage) was selected as the putative identity of that OTU. CSTs of vaginal microbiome were defined using Jensen–Shannon divergence and Ward linkage hierarchical clustering (*Gajer et al., 2012*). Scatter diagrams were generated on GraphPad Prism (GraphPad Software Inc., La Jolla, CA, USA).

Continuous variables (age, the number of pregnancies and α diversity indices) were statistically compared across groups using the Mann–Whitney Test, as these data were not normally distributed. Analysis of covariance (ANCOVA) was performed to adjust potential factors. Categorical variables (smoking status, vaginitis history) were presented as frequencies (percentages) and compared between groups by Fisher's Exact Test or chi-square test. All analyses were performed on SAS software (version 9.2). For all tests, a two-sided $P$ value less than or equal to 0.05 was deemed statistically significant.

# RESULTS

## Participants

Vaginal swabs were available from 60 women (23 FSWs and 37 FNSWs). Smoking rates between groups was not statistically different. However, the FSWs were younger than the FNSWs (Median 25 vs 29; $P = 0.001$), and had a greater number of pregnancies ($P = 0.012$). The proportion of FSWs with a history of vaginitis was higher than FNSWs ($P = 0.034$) (Table 1). By using RDA (Fig. 1A) and PERMANOVA analysis (Table S1), although the age and pregnancy history were associated with microbiota communities, they could only explain a less than 10% variance of microbiota community ($R^2 < 0.10$). The FNSWs were all Chinese, but FSWs included Vietnamese and Chinese women and the PERMANOVA analysis showed that the vaginal microbiome of FSWs with different nationalities was of no statistical significance ($R^2 = 0.062$, $P = 0.153$).

## FSWs have significantly different vaginal microbiota β diversity, but undifferentiated α diversity when compared with FNSWs

Although the ACE, Chao1 and Shannon indices for FSWs were higher than FNSWs, the differences were not statistically significant ($P > 0.05$). After adjusting for age, the number of pregnancies and vaginitis history, the differences of ACE, Chao1 and Shannon indices between FSWs and FNSWs were not statistical significant ($P > 0.05$; Table S2). Binary jaccard dissimilarity PCoA analysis showed that microbiota communities in FSWs were visibly different from FNSWs (Fig. 1B). The $R^2$ was 0.321 based on the PERMANOVA Test (Table S1). The cluster heatmap showed FNSWs were mostly clustered in right and FSWs were in left (Fig. 1C), which suggested that the vaginal microbiome structures of FNSWs and FSWs could be distinguished to some extent. The bacteria genera list in the vertical axis were the common genera of which relative abundances were more than 0.1%.

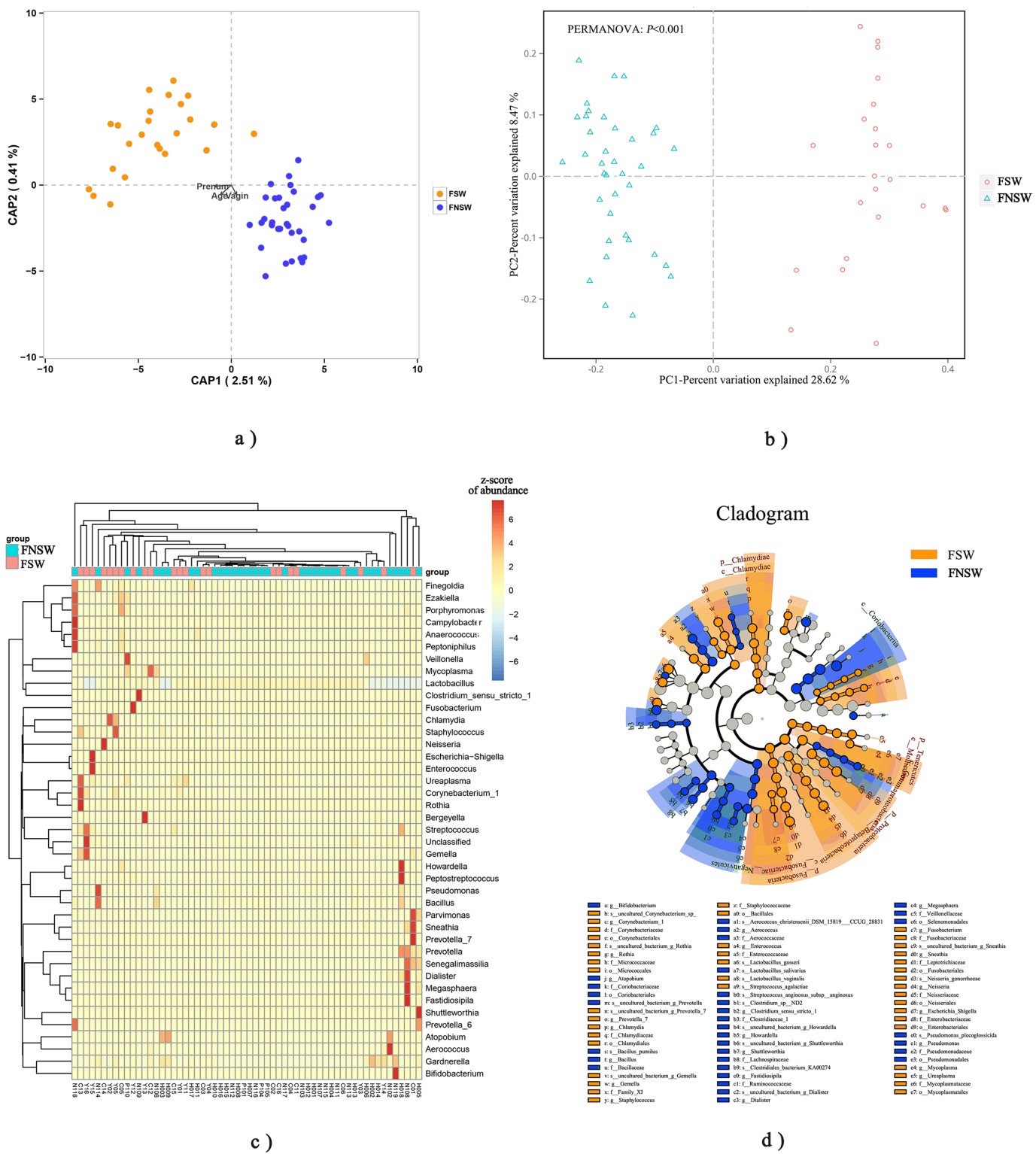

**Figure 1 Microbiota community diversity from female sex workers and non-sex workers.** (A) RDA analysis for correlations between microbiota diversity and age, pregnancy history and vaginitis history of participants; (B) Binary_jaccard dissimilarity PCoA analysis for the FSWs and FNSWs groups; (C) Cluster heatmap for different samples and genus; (D) Lefse analysis for the microbiota between FSW and FNSW groups, threshold LDA score = 2; FSW, female sex worker; FNSW, female non-sex worker.

**Table 1 Characteristics of female sex workers and female non-sex workers.**

|  | Female sex workers N = 23 | Female non-sex workers N = 37 | Z/χ² | P |
|---|---|---|---|---|
| Age (Median, Range), Year | 25 (18–41) | 29(20–40) | –3.368 | 0.001 |
| Smoke (Yes), n (%) | 5(21.7) | 8(21.6) |  | 0.999* |
| Pregnancy history (Median, Range) | 1(0–5) | 1(0–3) | –2.513 | 0.012 |
| Unknown | 0 | 3 |  |  |
| Vaginitis history (Yes), n (%) | 11(47.8) | 8(21.6) | 4.501 | 0.034 |
| Nationality |  |  |  |  |
| Chinese | 12 | 37 | – | – |
| Vietnamese | 11 | 0 |  |  |

**Note:**
  * Fisher's Exact Test.

## Specific genus comparison between FSWs and FNSWs

Lefse analysis (Fig. 1D) showed differences in microbiota orders, families, genera and species between the two groups. However only five genera reached the threshold LDA score of more than 3.5: FSWs with a higher abundance of *Chlamydia*, *Shigella*, *Neisseria* and *Sneathia* and FNSWs with a higher abundance of *Atopobium* (Fig. 2A).

The MeanDecreaseGini index from the random forest model (Fig. 2B) illustrated that *Atopobium*, *Neisseria* and *Enterococcus* were the key genera in distinguishing FSWs' and FNSWs' vaginal microbiota communities. To compare genus differences between these two groups, a multivariate Kruskal–Wallis test was performed (Fig. 2C). By FDR adjusting, the results were almost consistent with the results from the Lefse analysis. The average relative abundances of *Sneathia*, *Shigella*, *Neisseria*, *Chlamydia*, *Prevotella*, *Enterococcus* and *Ureaplasma* in FSWs was higher than those in FNSWs. The *Atopobium abundance* in FSWs was lower than FNSWs ($P < 0.05$). The scatter diagrams showed that a FSW sample was abnormally with high abundance of *Escherichia_Shigella* (Fig. 3C). No *Neisseria* was detected in FNSW samples but was positive in 21(91.3%) FSWs, and *Chlamydia* was detected positive in 18(78.3%) FSWs but only one in FNSW samples (Figs. 3D and 3E).

## The relative abundance of *Lactobacillus* spp. among FSWs and FNSWs

The *Lactobacillus* genus was the dominant species in most vaginal microbiota communities (37/60); however the average relative abundance between FSWs and FNSWs was not statistically significant ($P = 0.373$) (Fig. 4A). With regards to specific *Lactobacillus* strains, the relative abundance of *L. crispatus*, *L. gasseri* and *L. jensenii* in FSWs was lower than for FNSWs, and *L. iners* abundance was indistinguishable between two groups (Figs. 4B–4F). According to the specific dominant genus, the vaginal microbiota communities were divided into five types: CSTI (*L. crispatus*), CSTII (*L. gasseri*), CSTIII (*L. iners*), CSTIV (highly diverse dominant) and CSTV (*L. jensenii*). No CST V microbiota were observed in our 60 samples. CST distributions were not statistically different between FSWs and FNSWs ($P > 0.05$) (Table S3).

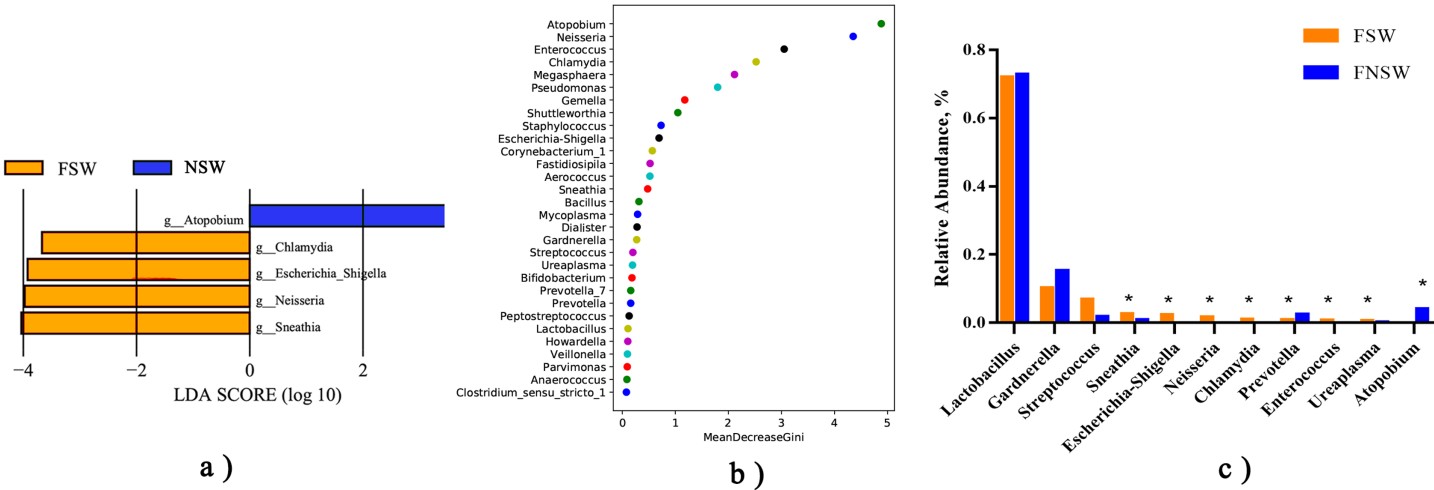

**Figure 2 The specific genus between female sex workers and non-sex workers.** (A) Lefse analysis for genus levels between FSWs and FNSWs, LDA score > 3.5; (B) MeanDecreaseGini index from random forest model; (C) The average relative abundance of specific genera between FSWs and FNSWs; the asterisk relates to the *P* value of the multivariate Kruskal–Wallis test was less than 0.05 by FDR adjustment. FSW, female sex worker; FNSW, non-sex worker.

### Influencing factors on FSW vaginal microbiota communities

Within the FSW group, RDA analysis revealed that correlations between vaginal microbiota communities and the average number of clients per week, drinking habits and duration of sex working (Fig. S1A) were relatively obvious, but with no statistical significance through PCoA and PERMANOVA analyses ($P > 0.05$) (Figs. S1B–S1D).

## DISCUSSION

To our knowledge, this is the first study that has focused on the vaginal microbiome of cross-border FSWs in southwest China. We found distinct differences in vaginal bacterial structures, but not diversities between FSWs and FNSWs. The average relative abundances of *Sneathia*, *Shigella*, *Neisseria*, *Chlamydia*, *Prevotella*, *Enterococcus* and *Ureaplasma* among FSWs was higher than for FNSWs. *Lactobacillus* was the major genus in both FSWs and FNSWs; no differences were observed between them for both groups. However, this was inconsistent with findings by *Wessels et al. (2017)*; the most likely reasons were the race, geography, and hygiene practices. But at a species level, *L. crispatus*, *L. gasseri* and *L. jensenii* in the FSW group were lower in abundance than the FNSW group. This study filled the gap of characteristics of the vaginal microbiome in cross-border FSWs in China.

The public health implications of this study are in two aspects as follows. Firstly, the data about FSWs, especially in cross-border areas, was valuable because the cross-border sexual work industry in China is illegal. In order to ensure the quality of investigation, this program was conducted by Chinese CDC whose remit is to control HIV transmission (*Wang et al., 2012*), and the samples were collected by standardized process. What we found could provide data for further research in this special group. Secondly, most FSWs here were very young or even juveniles, acquiring STIs or BV or vaginal dysbiosis will affect them going forward. Because most FSWs will go home to get married, we have reasons to believe that the negative effects of vaginal dysbiosis may impact their future

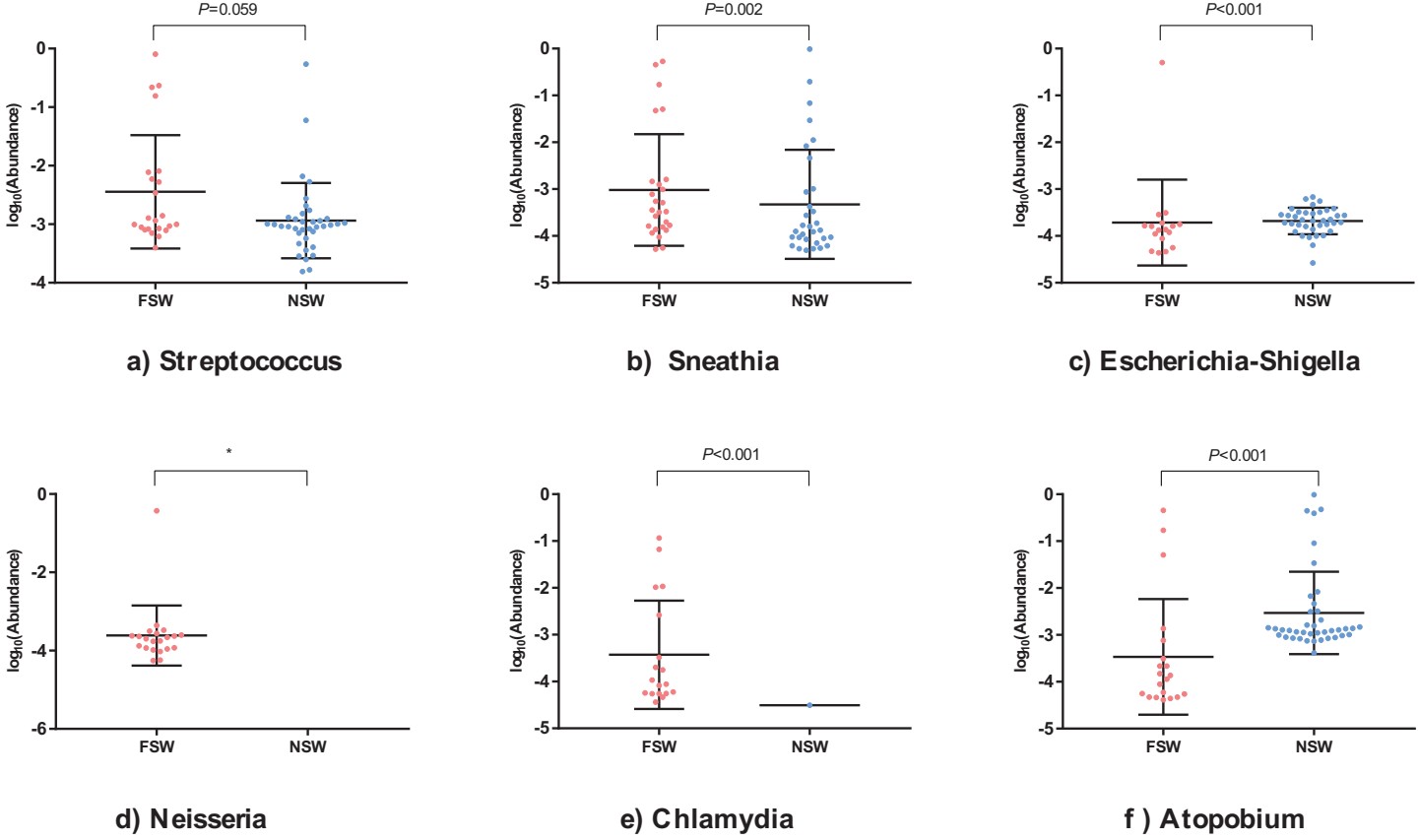

**Figure 3 The scatter diagrams of specific genera between female sex workers and female non-sex workers.** The relative abundance of (A) *Streptococcus*, (B) *Sneathia*, (C) *Escherichia Shigella*, (D) *Neisseria*, (E) *Chlamydia*, (F) *Atopobium* genus between groups. The *P* value was from the multivariate Kruskal–Wallis test and adjusted by FDR. FSW, female sex worker; FNSW, female non-sex worker.

family, such as STI transmission to husbands. The risk of infertility (*Campisciano et al., 2017*) or adverse pregnancy outcomes (abortion, preterm births and premature rupture of fetal membranes) (*Dunlop et al., 2015*; *Stout et al., 2017*) may be higher because of the disorganized vaginal microbiomes.

In general, people with multiple sexual partners are at higher risk for STIs and infection with associated pathogens (*Cabecinha et al., 2017*), such as *Treponema pallidum*, *Trichomonad*, *Neisseria* and *Chlamydia*. Our study also expectedly found a higher abundance of *Neisseria* and *Chlamydia* among FSWs. It should be noted that these STIs would potentially impact the vaginal microbiome, and an unbalanced vaginal microbiome would increase the risk of pathogen infection (*Ziklo et al., 2018*). We could not illuminate whether the different diversity of vaginal microbiome between FSWs and FNSWs was from the infection of *Neisseria/Chlamydia*. But in consideration of the high prevalence of STIs, if we made a strict standard to exclude them, the eligible FSWs would be limited and some unpredictable selection biases might exist. Moreover, the changes of relative abundance of *Neisseria* and *Chlamydia* were a typical performance of microbiota changes, and the complex correlations between the vaginal species were not explained clearly, so

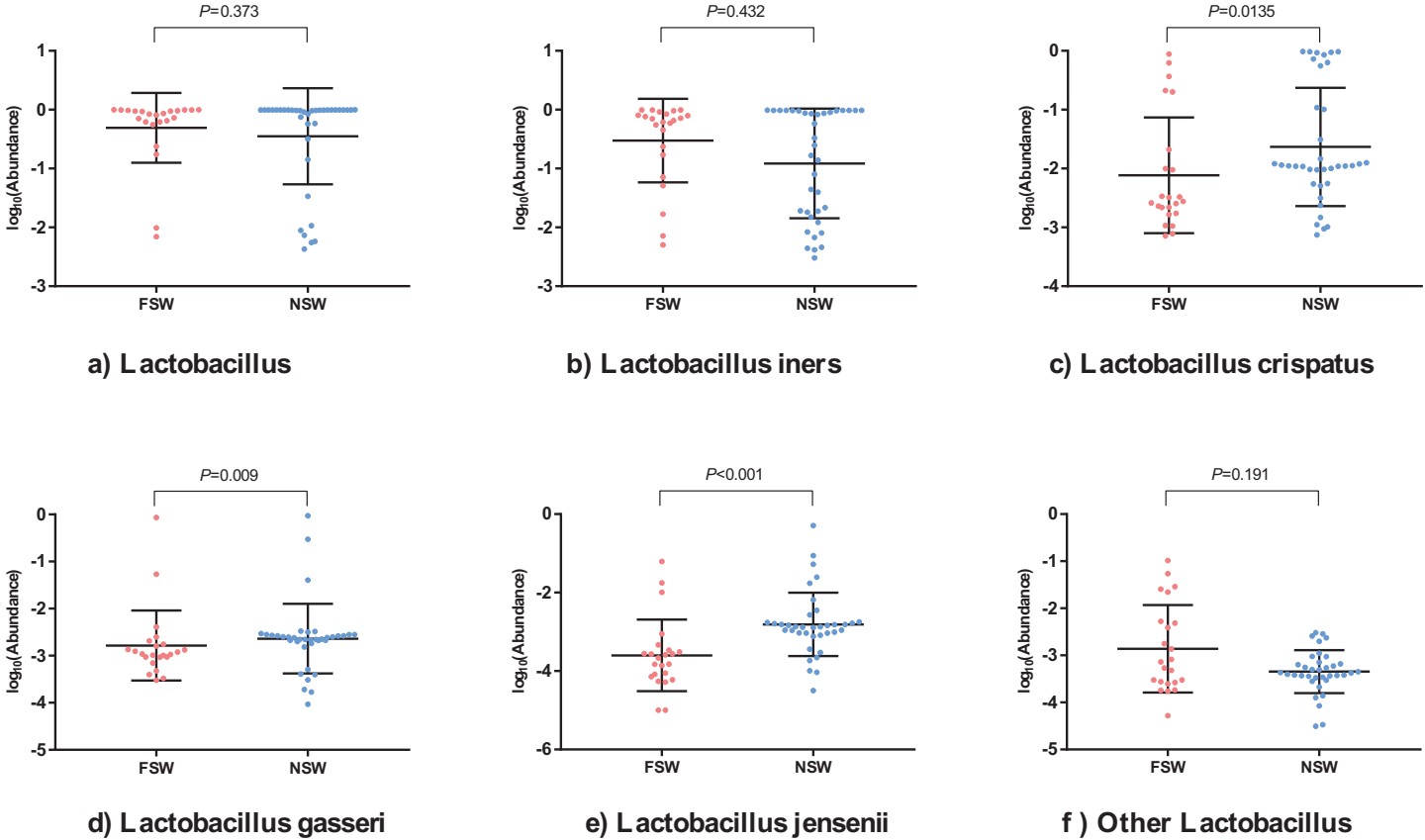

**Figure 4 The scatter diagrams of specific species between female sex workers and female non-sex workers.** The relative abundance of (A) *Lactobacillus genus*; (B) *Lactobacillus iners*; (C) *Lactobacillus crispatus*; (D) *Lactobacillus gasseri*; (E) *Lactobacillus jensenii*; (F) Other *Lactobacillus* species between two groups. The *P* value was from the Kruskal–Wallis test. FSW, female sex worker; FNSW, female non-sex worker.

exploring the whole vaginal microbiota communities which were not specially selected were valuable. Meanwhile, the increased relative abundance of *Sneathia*, *Streptococcus* and *Shigella* among FSWs were observed. *Sneathia* is a Gram-negative, rod-shaped, non-spore-forming and non-motile bacteria; it has the potential to impact on bacterial vaginosis and cause preterm births (*Harwich et al., 2012*). *Streptococcus*, especially the Group B *Streptococcus* (GBS), is always detected in the female genital tract, and is an important pathogen causing adverse pregnancy outcomes and neonatal infections. *Rosen et al. (2017)* observed that vaginal microbiota CST and α-diversity are not related to GBS status, however, specific microbial taxa are associated with the colonization of this important human pathogen. *Shigella* is a pathogenic enterococci typically located in the intestinal tract. When it invades the vagina, the pathogen causes severe vaginitis symptoms, including bloody vaginal discharge (*Murphy & Nelson, 1979*). It should be noted that anal sex behaviors were common among FSWs (*Kelly-Hanku et al., 2014*), which may explain the increasing of entero-associated bacterial infections. *Atopobium* could be detected in healthy women, but its role was controversial. It is commonly associated with bacterial vaginosis and an increasing risk of adverse pregnancy outcomes (*Mendes-Soares et al.,*
2015). But genetic differences across *Atopobium* species suggest that single species could be associated with both health and disease (*Mendes-Soares et al., 2015*). In the FNSW group, the relative abundance of the *Atopobium* genus was higher, but the specific species was unknown. In the future, metagenomic sequencing applications could identify the *Atopobium* species.

Although there were no statistical significance of relative abundance of *Lactobacillus* genus among FSWs and FNSWs for our small sample size, the specific species of *Lactobacillus* were different between groups. *L. crispatus* in FNSWs was higher than in FSWs, which plays a crucial role in protecting the genitourinary tract against pathological conditions. *Rizzo et al. (2015)* found that *L. crispatus* reduced IL-6, IL-8 and TNF-α production, but specifically enhanced IL-10 anti-inflammatory cytokines in *Chlamydia trachomatis*-infected HeLa and J774 cells, suggesting specific *L. crispatus* protective mechanisms at the vagina. However, the effect of *L. iners* on vaginal health was unclear. Biochemical and functional assays have suggested that *L. iners* contain features of probiotic *lactobacilli* as well as vaginal pathogens, which may be based on genome clonal variants (*Petrova et al., 2017*). Some studies have shown that *L. iners* dominated vaginal microbiome community appear to be less stable or more in transition than other community types and are associated with vaginal dysbiosis (*Verstraelen et al., 2009*). Our results revealed that the average relative abundance of *L. iners* among FSWs was higher than for FNSWs, although the differences were not statistically significant. Similarly, the abundance of *L. gasseri* among FSWs was lower than for FNSWs, which was consistent with *De Backer et al.'s (2007)* study. The negative association between *L. iners* and *L. gasseri* further suggested that bacteria in vaginal communities are interrelated.

At present, the CST is the recognized classification method (*Gajer et al., 2012*), with each type characterized by a specific and typical composition and abundance of taxa. Most vaginal communities and corresponding CSTs are dominated by one or several *Lactobacillus* species (*Ravel et al., 2011*), referred to as CSTI (*L. crispatus*), CSTII (*L. gasseri*), CSTIII (*L. iners*), CSTIV (*high diversity dominant*) and CSTV (*L. jensenii*). The associations between different CSTs and health had been studied in other populations (*Campisciano et al., 2017*; *Chen et al., 2017*; *Rosen et al., 2017*; *Wessels et al., 2017*), but results have been inconsistent. In our analysis, we used this classification method to compare differences in CST composition between FSWs and FNSWs, however no meaningful differences were observed. Some researchers have suggested that vaginal microbiome stability is usually not expressed in terms of changes in taxa composition, but rather in terms of CST consistency (*Gajer et al., 2012*). Therefore, we should conduct future studies on associations between CST changes and health status, not just the current CST situation in one time.

The present study was limited due to its cross-sectional elements. The vaginal microbiome is ecologically dynamic (*Greenbaum et al., 2018*); therefore, going forward, the best approach is to explore the different structures of the vaginal microbiome between these two populations over at least one menstruation cycle (*Uchihashi et al., 2015*). Meanwhile, our cross-sectional study could not explain the causality between STIs and vaginal microbiome changes, and whether commercial sex behaviors led STIs first or

impact vaginal microbiome first. Given the number of factors that impact the vaginal microbiome, our data only scratches the surface. For instance, lubricant use and menstruation data among FSWs, and FNSW sexual behaviors were not collected. Equally, information bias was inevitable because of our sensitive questions. In addition, the small sample size may have reduced adequate cohort representation, although our samples were randomly selected, and the inclusion criteria were strict. Further, because of the specific characteristics of FSWs, our FSW participants were younger and had higher pregnancy and vaginitis histories, which affected comparability within the groups. Meanwhile, our control group only included the Chinese FNSWs and not Vietnamese FNSWs, which was one of the sources of heterogeneity between FSWs and FNSWs. Our results should be cautiously extrapolated to FSWs in other jurisdictions, because cross-border FSWs in China have different lives and sexual features.

## CONCLUSIONS

Our results have indicated distinct differences in vaginal bacterial structures between FSWs and FNSWs; some disease-related genera, such as *Sneathia*, *Shigella*, *Neisseria* and *Chlamydia*, were more abundant in FSWs. Because of the important health effects of the vaginal microbiome and the high transience of FSWs, further research to identify specific types of microbiome structures related to FSWs or STI risks is warranted.

## ACKNOWLEDGEMENTS

We thank the staff at CDC in Hekou County for their work on data collection and International Science Editing for English language editing.

### Funding

This work was supported by the National Natural Science Foundation of China (No. 81872634), the Fundamental Research Funds for the Central Universities (2242016K40025) and the Postgraduate Research and Practice Innovation Program of Jiangsu Province (KYCX17_0184). The funders had no role in study design, data collection and analysis, decision to publish, or preparation of the manuscript.

### Grant Disclosures

The following grant information was disclosed by the authors:
National Natural Science Foundation of China: 81872634.
Fundamental Research Funds for the Central Universities: 2242016K40025.
Postgraduate Research and Practice Innovation Program of Jiangsu Province: KYCX17_0184.

### Competing Interests

The authors declare that they have no competing interests.

## Author Contributions

- Xiang Hong conceived and designed the experiments, performed the experiments, analyzed the data, contributed reagents/materials/analysis tools, prepared figures and/or tables, authored or reviewed drafts of the paper, approved the final draft.
- Shenghao Fang analyzed the data, prepared figures and/or tables, authored or reviewed drafts of the paper, approved the final draft.
- Kaiping Huang performed the experiments, authored or reviewed drafts of the paper, approved the final draft.
- Jiechen Yin conceived and designed the experiments, analyzed the data, contributed reagents/materials/analysis tools, prepared figures and/or tables, approved the final draft.
- Jianshuang Chen performed the experiments, authored or reviewed drafts of the paper, approved the final draft.
- Yan Xuan performed the experiments, authored or reviewed drafts of the paper, approved the final draft.
- Jing Zhu authored or reviewed drafts of the paper, approved the final draft, sample collection.
- Jun Ma performed the experiments, contributed reagents/materials/analysis tools, prepared figures and/or tables, approved the final draft.
- Pengfei Qin authored or reviewed drafts of the paper, approved the final draft, sample collection.
- Danhong Peng authored or reviewed drafts of the paper, approved the final draft, sample collection.
- Ning Wang authored or reviewed drafts of the paper, approved the final draft, sample collection.
- Bei Wang conceived and designed the experiments, authored or reviewed drafts of the paper, approved the final draft.

## Human Ethics

The following information was supplied relating to ethical approvals (i.e., approving body and any reference numbers):

The Chinese center for disease control and prevention granted Ethical approval to carry out the study (Ethical Application Ref: X120331209).

## Data Availability

The sequences are available at Figshare: Hong; Bei, Wang (2019): vaginal microbiome 16S rDNA sequencing data. figshare. Dataset. DOI 10.6084/m9.figshare.8479301.v1.

The original sequencing data are available at SRA: PRJNA551362.

## Supplemental Information

Supplemental information for this article can be found online at http://dx.doi.org/10.7717/peerj.8131#supplemental-information.

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
