# Peer review of "Characteristics of the vaginal microbiome in cross-border female sex workers in China: a case-control study"

_PeerJ, doi:10.7717/peerj.8131_

## Round 0.1 · original submission · Major Revisions

When responding to the comments of the two reviewers, please pay particular attention to the discussion regarding possible associations with Neisseria and Chlamydia. It may be difficult to completely resolve this issue given the available subjects and the number of samples, but a more thorough discussion of the issue is in order. In a similar manner, any association with country of origin (Vietnam or China), FSWs and FNSWs, and the microbiome should be further explored and discussed.

Reviewer 1 ·

Basic reporting

1.Title: Usually reported as a “case-control study”, rather than case controlled
Abstract Background: I think the implication is that understanding differences in the vaginal microbiome is important for STI control among all women; the authors selected sex workers as an extreme group to look for variations that might be associated with STIs. This could be made clearer.
Introduction: Please do not start off with a limitation. The second sentence is much more engaging and the illegal status of sex work can be addressed in the discussion.
Thank you for acknowledging that sex work, especially trans-border sex work, is a difficult social issue and not problematizing the women who engage in sex work.
Line 60: Lactobacillus is a broad genus; some species have been found to be more protective and indeed produce hydrogen peroxide. Others however, in particular L iners, are less associated with vaginal health. Please update your review in lines 59-63.
Line 63: “the mucosal barrier of the vagina was more easily broken” or disrupted
Line 66: Unclear what “except for pathogen infection” means; also unclear what reproduction associated diseases. Are you explaining that dysbiosis can include BV and is associated with negative pregnancy-related outcomes?
Line 72: The number of community state types varies greatly across the literature; the 5 state model is an appropriate choice, but should be indicated as a choice. The lactobacillus discussion here is much more appropriate and perhaps that earlier section can be included here?
Line 81: The critique of the African study seems unnecessary, as you are not addressing that particular question.
Line 90: Vaginal swabs (cut “information”)
Line 99: For those not familiar with the geography, how close are these two study sites?
Line 100: Please capitalized Chinese Center for Disease Control consistently.
Line 148: p-value should be less than 0.05 to be significant
Line 168: Please provide more explanation than clustered in the right or clustered in the left.
Line 194: Unclear what “divided by” refers to?
line 203: Race, geography, and hygiene practices could all explain these differences.
line 205: This is a new and somewhat unexpected hypothesis – understanding how FSW impact the COMMUNITY microbiome goes far beyond the sampling and methodology of this paper.
line 206: This paragraph needs to be edited, both to improve the language (grammar and typos), but to have a clearer purpose. Are you outlining the strengths and limitations of the study? If so, please state this clearly.
line 218: Please be clearer: Neisseria and Chlamydia are known pathogens and it is not surprising they were found more often among FSW; it is unclear that they alter pH. Overall however, this paragraph is good and provides interpretation and meaning to the results.
Line 266: This is an excellent summary of limitations but should be incorporated into the paragraph starting at line 206. The ideas for future research are good.
Figure 1a: what do the colors and shapes indicate?
Figure 1b: also address in the text, what species loaded onto each principal component?
figure 1c: what does the scale -6 to 6 refer to?
Please include somewhere the proportion of each group with Neisseria and Chlamydia; antibiotic use and the other factors described in the limitations should also be included in Table 1.
Please present the CST by groups.

Experimental design

Abstract results: It is unusual to include STIs themselves in the analysis (Neisseria and Chlamydia), especially as the premise is to look for other vaginal microbes that might be associated with acquisition of Neisseria and Chlamydia.
Line 85: As there are no approved treatments to successfully change the vaginal microbiome, the second part of this sentence seems inappropriate
Methods: It is interesting that you describe community state types in the introduction but do not use them to characterize your samples.
Line 186: Community types are used! Please include this in your methods section as well. The results are very interesting and it is valuable that you were able to examine so many lacto species.
Power: How was the sample size determined? How were the control samples selected – was there any matching on age, ethnicity, etc?
Please state a specific hypothesis you are testing, for example – we hypothesized that FSW would have higher alpha and beta diversity relative to NFSW.

Validity of the findings

Are any of the differences in diversity or PCA still present when Neisseria and Chlamydia are removed? If your hypothesis is that differences in microbiome lead to Neisseria and Chlamydia infection, it is inappropriate to include these bacteria in your analysis. It would be tempting to look for associations between microbiome and Neisseria/Chlamydia among the sex worker samples – but your sample size may be too small.
Line 239: Unclear what you are saying. There are clear shifts in CST between your two groups (in the supplemental materials), even if they do not meet statistical significance (give the small numbers, Fishers Exact might be a better test). How are you defining dysbiosis? I don’t see this anywhere in the results.
line 243: Why would hospital patients be more likely to have vaginal dybiosis?
Conclusion: I would rethink what these results mean. What would the benefit of screening actually be?
Figure 3. Unless you also performed quantitative PCR, I’m unclear that log(Abundance) is a meaningful measure.

Additional comments

Thank you for undertaking this work; I believe the data have potential to be very informative. However, I think the analysis must be updated to exclude Neisseria and Chlamydia and then re-assess your results. In addition, please review appropriate methods for comparing relative abundances of specific species. Finally, I am concerned about your sample size.

Reviewer 2 ·

Basic reporting

This is an interesting article describing differences found in the vaginal microbiota of female sex workers in southwest China. The article is well presented and has interesting findings, however there are several places where the English is not up to standard and needs to be revised:
1. Lines 62-63 are very poorly worded and I can't understand the meaning. Revision is needed.
2. Lines 81-82 are also poorly worded.
3. Line 118 "Hisq" should be "HiSeq"
4. Line 137 "Line" should be "Linear"
5. Line 175 "this" should be "these"
6. Lines 183 and 184 "related" should be "relative"

Experimental design

The use of 97% OTUs in the primary analysis of this data would likely obscure several different species of Lactobacillus which have representative sequences that get grouped together into the same OTU. Use of 99% OTUs would be far more accurate in this type of study, or better yet use of a more advanced method like DADA2 which isolates biologically meaningful sequence variants.

Validity of the findings

The findings of differences between FNSW and FSW are quite stark and particularly the figure panel b showing PC1 vs PC2 is a perfect separation of FSW and FNSW. This pronounced of a difference leads me to believe (speculate) that the differences the authors are seeing may be mainly driven by the fact that the FSW's are primarily of Vietnamese descent (crossing into the region at the border) whereas they are being compared to the FNSW's who are likely primarily of Chinese descent. This is something that the authors do a good job of pointing out in the discussion, however it is not highlighted as a limitation in the abstract. Readers shouldn't have to dig for this important of a distinction. Further, Table 1 should include a breakdown of the background of the FSW's and FNSW's so that the reader can see how much of an issue this may be. How many of the FNSW's were of Chinese background vs Vietnamese? How many of the FSW's were of Vietnamese background vs Chinese?

---

## Round 0.2 · accepted · Accept

We are please to accept your manuscript for publication in PeerJ. As indicated by one of the reviewers, the grammar could still be improved a bit. Please work with PeerJ Production to tweak the English to improve grammar. Thank-you.

Reviewer 1 ·

Basic reporting

The authors have made good changes throughout that strengthen the article. There are still many places where grammar could be improved and I recommend one more round of edits to polish the language.

Experimental design

The authors addressed the questions raised well. The sensitivity analysis showing that not all the differences between FSW and FNSW were due to Chlamydia and Gonorrhea are important and should be included. Likewise, the analysis between Vietnamese and Chinese patients is a nice addition. The edits to the tables and figures are helpful.

Validity of the findings

The results are interpreted well and the limitations are appropriate.

Additional comments

Thank you for the thoughtful edits.

Reviewer 2 ·

Basic reporting

The authors have adequately addressed my concerns about better presentation of the manuscript via fixing of typos and correcting of english language deficiencies pointed out.

Experimental design

I did not have any issues with the original experimental design and the authors have addressed any concerns raised by the other reviewer adequately.

Validity of the findings

The authors have addressed the limitations that were pointed out and added the appropriate disclaimers to the findings. Particularly the issue that FNSW were all of Chinese background is a limitation of the experimental design and is now appropriately pointed out and detailed in the demographics.